# Anatomical variations of the equine femur and tibia using statistical shape modeling

**Hongjia He**[1]☯, **Scott A. Banks**[2]☯, **Adam H. Biedrzycki**[1]☯*

**1** Department of Large Animal Clinical Science, College of Veterinary Science, University of Florida, Gainesville, Florida, United States of America, **2** Department of Mechanical & Aerospace Engineering, Herbert Wertheim College of Engineering, University of Florida, Gainesville, Florida, United Stated of America

☯ These authors contributed equally to this work.
* dradam@ufl.edu

**Data Availability Statement:** Here is the link for the files used in this study: https://www.ebi.ac.uk/biostudies/studies/S-BSST1116.

**Funding:** HH The project was partially funded by the University of Florida Graduate Student

## Abstract

The objective of this study was to provide an overarching description of the inter-subject variability of the equine femur and tibia morphology using statistical shape modeling. Fifteen femora and fourteen tibiae were used for building the femur and tibia statistical shape models, respectively. Geometric variations in each mode were explained by biometrics measured on ±3 standard deviation instances generated by the shape models. Approximately 95% of shape variations within the population were described by 6 and 3 modes in the femur and tibia shape models, respectively. In the femur shape model, the first mode of variation was scaling, followed by notable variation in the femoral mechanical-anatomical angle and femoral neck angle in mode 2. Orientation of the femoral trochlear tubercle and femoral version angle were described in mode 3 and mode 4, respectively. In the tibia shape model, the main mode of variation was also scaling. In mode 2 and mode 3, the angles of the coronal tibial plateau and the medial and lateral caudal tibial slope were described, showing the lateral caudal tibial slope angle being significantly larger than the medial. The presented femur and tibia shape models with quantified biometrics, such as femoral version angle and posterior tibial slope, could serve as a baseline for future investigations on correlation between the equine stifle morphology and joint disorders due to altered biomechanics, as well as facilitate the development of novel surgical treatment and implant design. By generating instances matching patient-specific femorotibial joint anatomy with radiographs, the shape model could assist virtual surgical planning and provide clinicians with opportunities to practice on 3D printed models.

## Introduction

The equine stifle has been a comparatively overlooked joint from an advanced biomechanical perspective due to difficulties acquiring volumetric images in comparison to the distal limb joints. Equine stifle osteoarthritis (OA) is a common source of lameness and can be spontaneous or secondary to trauma [1–4]. In humans, numerous studies have investigated the role of joint morphology in the onset and progression of OA in order to understand the relationship between shape and the development of OA [5–7]. Joint geometry plays an essential role in

Fellowship Award. The funders had no role in study design, data collection and analysis, decision to publish, or preparation of the manuscript.

**Competing interests:** The authors have declared that no competing interests exist.

biomechanical behaviors and contributes to the progression of human knee OA [8]. However, the relationship between equine stifle morphology and OA has yet to be explored, and detailed descriptions about the anatomical shape variation in the equine femur and tibia are still lacking.

One common method to characterize complex anatomical geometry is statistical shape modeling (SSM), which has been employed to quantify the anatomical morphological variability in various joints in humans, such as carpometacarpal joint and patellofemoral joint [9, 10]. In horses, SSM has been used to evaluate the equine third metacarpal and also an articulating model of the equine distal limb [11, 12]. SSM creates a point-to-point-based correspondence within samples and extracts the most prominent modes of shape variation within the population, named principal components or modes, using principal component analysis (PCA), which mathematically reduces the shape dimensionality of the region of interest [13, 14]. Several studies utilized SSM to provide better understanding regarding the correlation between joint morphology and joint function, which enabled prognosis and determination of progression in humans for joint OA [15, 16]. With advanced imaging technologies such as computed tomography (CT) becoming more common in veterinary practice, equine SSM studies are now being published and in future could be used to guide therapies and evaluate risks based on morphologies [11, 12].

This study aimed to create statistical shape models of the equine femur and tibia to provide an overarching description of morphological variability in the equine stifle, which could serve as a baseline for future investigations on correlation between equine joint morphology and degenerative joint diseases such as equine stifle OA. The subjective of this study was to describe the shape variations of the equine femur and tibia using shape models within ±3 standard deviations (SD) in each mode, which is commonly used in SSM to capture 99.7% of the shape variation [9]. We hypothesize that the main variation within both femur and tibia models would be the scaling of the bones.

## Materials and methods

### Population

This study was approved by the Institutional Animal Care and Use Committee (IACUC) at the University of Florida (Approval Number 201809196, August 7th, 2019). Adult horses from the hospital and research populations were used for this prospective study. Horses were euthanized for reasons unrelated to any hindlimb injuries. Colic was the main reason for euthanasia for clinical animals, and owner permission was obtained prior to use in the current research project. Horses euthanized at the conclusion of their *in vivo* research studies were also utilized in this study. All animals were euthanized via intravenous lethal injection of pentobarbital sodium (Beuthanasia-D) at 0.2ml/kg. Adult horses from the hospital and research populations were used for this prospective study. Horses were euthanized for reasons unrelated to any hindlimb injuries. No restriction was placed on age, breed, sex or body mass for inclusion in this study, other than the animal was required to be an adult and fully mature, which was classified as greater than 5 years of age.

### Sample collection and imaging protocol

All hindlimbs used were transected at the coxofemoral joint; no preference was given to laterality and the decision to harvest a left or right limb was based on a coin toss. Once harvested, limbs were scanned within 24 hours of collection. Bone algorithm volumetric data Digital Imaging and Communications in Medicine (DICOM) images of all specimens, from the proximal femoral head at the coxofemoral joint to the distal tibia at the tibiotarsal joint, were created

using a 160-slice Toshiba Aquilion CT Scanner (slice thickness of 0.5 mm, and in-plane resolution of 0.3 mm). Once CT images were acquired, scans were reviewed to evaluate for any obvious pathology (e.g., subchondral bone cysts, osteophytes etc.) and excluded if any obvious pathology was present.

## Statistical shape modeling

Segmentation for the femora and tibiae from DICOM data were performed in medical imaging software (Mimics 16.0, Materialise, Leuven, Belgium). Images were segmented using the program's predefined bone threshold (i.e., 226 to 3017 Hounsfield units). The segmented femora and tibiae were then exported as 3D objects represented by triangular meshes into 3-Matic (Materialise, Leuven, Belgium). All right limbs were mirrored to the left to create the SSM model. All samples underwent minimal preprocessing, such as smoothing, to eliminate noise from the CT images while the native anatomic geometry was mostly preserved. Briefly, the 3D samples were processed with smoothing and wrap functions using a gap closing distance and minimal defect size of 0.2 mm. The original femur and tibia samples were represented by approximately 55000 to 95000 and 45000 to 75000 triangles of various sizes, respectively. To achieve better results of shape registration with scaling and fitting as well as minimizing the computational expense, femur and tibia samples were remeshed with respective uniform triangles of 4.5 mm and 1.5 mm average edge lengths, which enabled a reduction of triangles to approximately 14000 to 18000 and 11000 to 14000 for femora and tibiae, respectively. For each model, one sample was arbitrarily chosen to be the template for the first iteration of point correspondence registration. Alignment with scaling of the samples were carried out with each sample being automatically rigidly aligned to the template and scaled. To minimize the computational expense while maintaining good mesh fidelity, sample points were set at 75% with 50 iterations for each alignment. A warp function was used to create point correspondence between the template and the sample after the alignment. Procrustes alignment was carried out to eliminate the position differences and ensure only shape variations were captured. Two separate PCAs were carried out to generate mean shapes for the femur and tibia models as well as to extract all modes of variation. The number of modes obtained was limited to n-1, where n was the number of samples used for each model building [17]. To minimize the potential error resulting from the arbitrarily chosen template, the mean model generated by the first iteration of the statistical model was used as the new template for the subsequent shape model building (Fig 1). The same process was followed, and a leave-one-out analysis was performed to evaluate the generalizability of each model. All principal components were named "modes" in the current study, for instance, the first principal component of the equine femur shape model would be mode 1 of the femur model.

## Shape analysis

Based on anatomical landmarks, measurements were performed on the femur and tibia instances generated by the respective shape model within ±3SD (i.e., mean, ±1SD, ±2SD and ±3SD instances) in 3-Matic. A cumulative variation of 95% within the population was used to determine the number of modes in the shape models to be included. Shape analysis with detailed measurements were performed on regions of interest as outcome measures: full length of the bone, condylar dimensions, femoral head size and orientation, femoral trochlear tubercle orientation, anatomical and mechanical axes orientation, femoral version angle, tibial plateau angles in the coronal and sagittal planes.

**Femur biometrics.** For the femur shape model, the length of the femur was measured from the most proximal point at the greater trochanter to the most distal point at the lateral

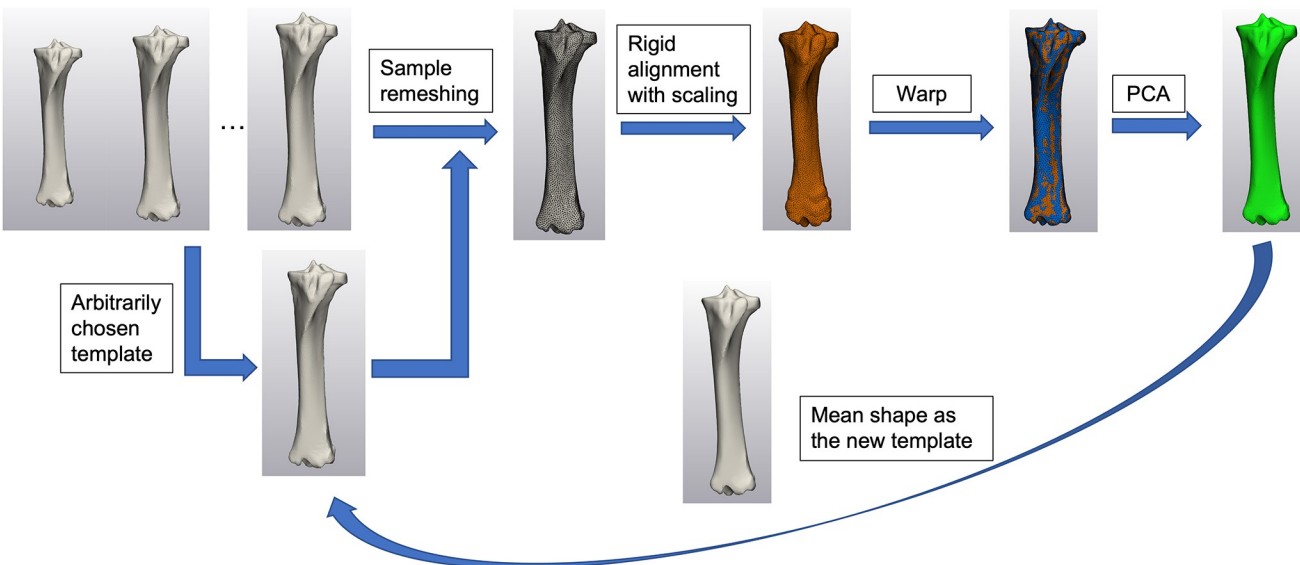

**Fig 1. Pipeline to generate statistical shape model of the tibia from segmented 3D samples exported from Mimics.** Image with blue and orange bones overlapping demonstrates non-rigid alignment between the template and the samples. The tibia statistical shape model (in green) created using the PCA generated mean model, was then used in the second iteration of SSM as a template.

tubercle of the femoral trochlea in the sagittal plane within mode 1 (Fig 2A). The bicondylar width was measured in mode 1 from the utmost medial point to the utmost lateral point on the femoral condyles in a transverse view (Fig 2B). The medial condylar width and depth were taken from a caudal view and a medial view of the femur in mode 1, respectively (Fig 2C and 2D). A spherical approximation to the medial femoral condyle was created in mode 1 using the condylar surface, and the radius of the best-fit sphere was used as measurement for the femoral condylar curve (Fig 2E). Similarly, a spherical approximation to the femoral head surface was created to quantify the radius of the femoral head within mode 1 instances (Fig 2F). The anatomical axis was created using the surface of the femoral diaphysis. The orientation of the femoral neck was the angle between the femoral head direction, which was defined by using a best-fit function to the femoral neck surface, and the anatomical axis measured in mode 2 (Fig 3A) [18]. The center of the femoral condyles was determined to be the midpoint of the centers of the best-fit spheres to the medial and lateral condyles. The mechanical axis was determined by the direction of the line connecting the center of the femoral head and the center of the femoral condyles, and the femoral mechanical-anatomical (FMA) angle was analyzed using ±3SD instances in the coronal plane in mode 2 (Fig 3B) [19, 20]. To facilitate describing the morphology of the medial tubercle of the femoral trochlea demonstrated by the shape model, a best-fit sphere to the most prominent region of the medial tubercle was created on instances generated in mode 3. The estimated direction of the tubercle follows the line connecting the center of the femoral condyles and the center of the best-fit sphere of the tubercle (Fig 4A). In a transverse plane that is normal to the anatomical axis, the femoral version angle was measured between the axis of the femoral neck and the transcondylar axis on instances generated in mode 4 (Fig 4B) [21].

**Tibia biometrics.** In the tibia shape model, the length was measured in mode 1 from the most proximal point on the medial intercondylar eminence of the tibia (MICET) to the most distal point of the distal intermediate ridge of the tibia in the coronal view (Fig 5A). The bicondylar width was measured from the most medial point to the most lateral point on the

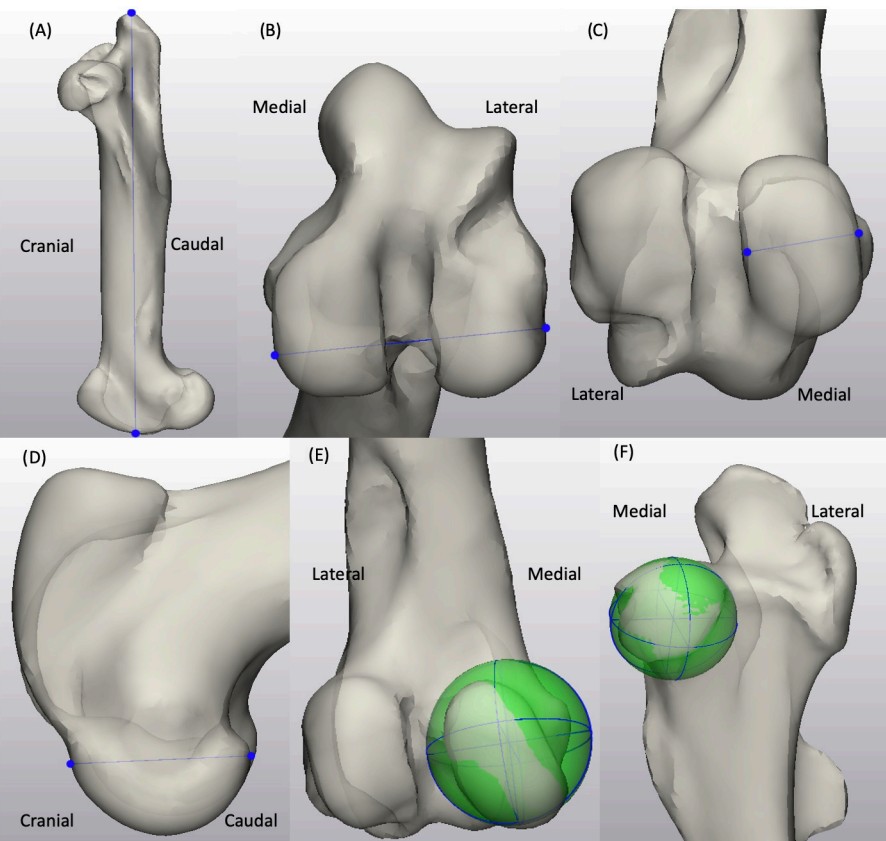

**Fig 2. Biometrics demonstrated on mode 1 instances generated by the femur SSM model: (2A) length of the femur, (2B) femoral bicondylar width, (2C, 2D) medial femoral condylar width and depth, (2E) estimation of the medial femoral condyle radius, and (2F) the femoral head radius.**

periphery of the tibial plateau in a transverse plane on mode 1 generated instances (Fig 5B). From proximal to distal in a transverse view, the medial condylar width was measured from the most medial point of the tibial plateau to the border of the MICET, which was characterized by a curvature analysis performed in 3-Matic (Fig 5C). The medial condylar depth was the distance from the most cranial point to the most caudal point on the tibial plateau measured in the sagittal plane on mode 1 instances (Fig 5D). The coronal tibial plateau (CTP) angle, represented by the joint line connecting the most medial and most lateral point on the tibial plateau and the anatomical axis, was analyzed in a cranial view on instances generated in mode 2 (Fig 6A) [22]. The medial and lateral caudal tibial slope (CTS) angles were measured in mode 3, the slope was defined by the angle between the line perpendicular to the anatomical axis of the tibial diaphysis and the line connecting the most cranial and caudal point on the tibial plateau in the sagittal plane (Fig 6B) [23].

## Statistics

To evaluate the correlations between the observed biometrics and the corresponding mode of variation, correlation analyses using Pearson's correlation coefficient R with statistical significance set at $p < 0.05$ were performed using 100 shape model generated femur instances with 4 modes at arbitrary standard deviations. The same analyses were also performed on 100 tibia instances with 3 modes at randomly generated standard deviations. In addition, a Pearson's

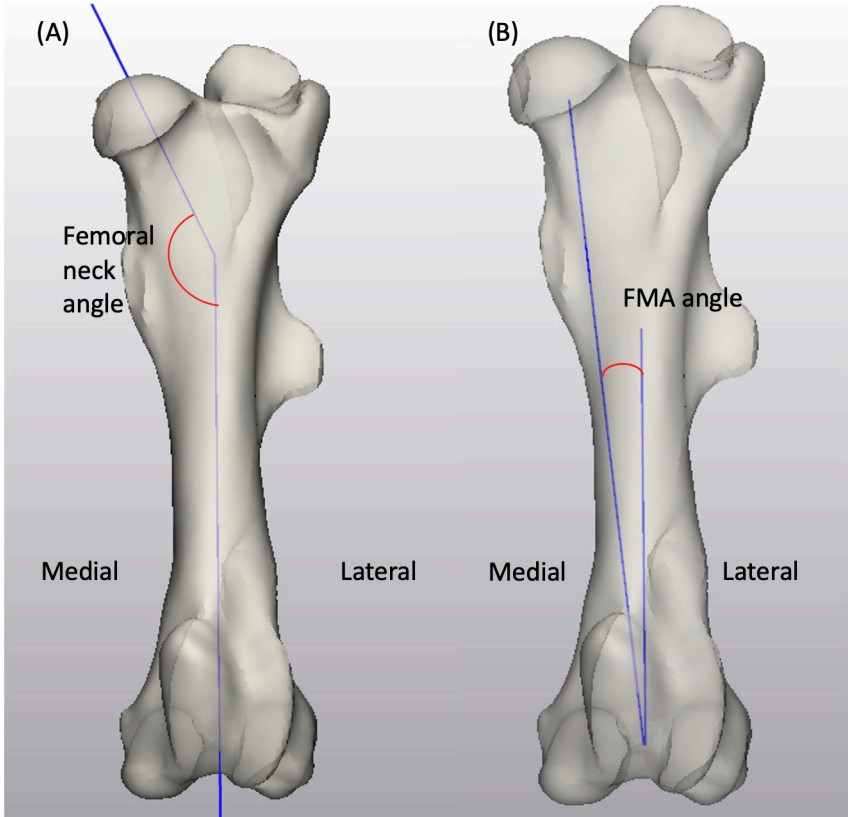

**Fig 3. Biometrics demonstrated on mode 2 instances generated by the femur SSM model: (3A) femoral neck angle and (3B) FMA angle.**

correlation analysis and a paired T test with statistical significance set at $p < 0.05$ was performed to evaluate the relationships between the CTS angle in the medial and lateral compartments.

## Results

Fifteen equine hindlimbs with appropriate institutional approval were used in the study. One Morgan horse, two Quarter Horses and twelve Thoroughbred (9 females, 1 stallion and 5 geldings; mean age ± SD = 14.3 ± 4.6 years old; weighing 488.7 ± 46.8 kg) were used. Only entire femora or entire tibiae were used; one tibia that was sectioned at the mid-diaphysis was excluded, thus fifteen femora and fourteen tibiae were used. All specimens were examined through CT images and displayed no signs of injury or obvious deformation in the stifle joint.

### Model evaluation

Cumulative variance accounted for by ordered modes decreases for both femur and tibia statistical shape models (Fig 7). With 6 and 3 modes, approximately 95% of the total geometric variations in the sampled population were described for the femur and tibia shape models, respectively. The leave-one-out analysis for both shape models showed a decrease in root-mean-square error as the number of modes used in the prediction model increases (Fig 8). The average geometric error for femur and tibia shape models were 2.12 mm and 1.37 mm, respectively.

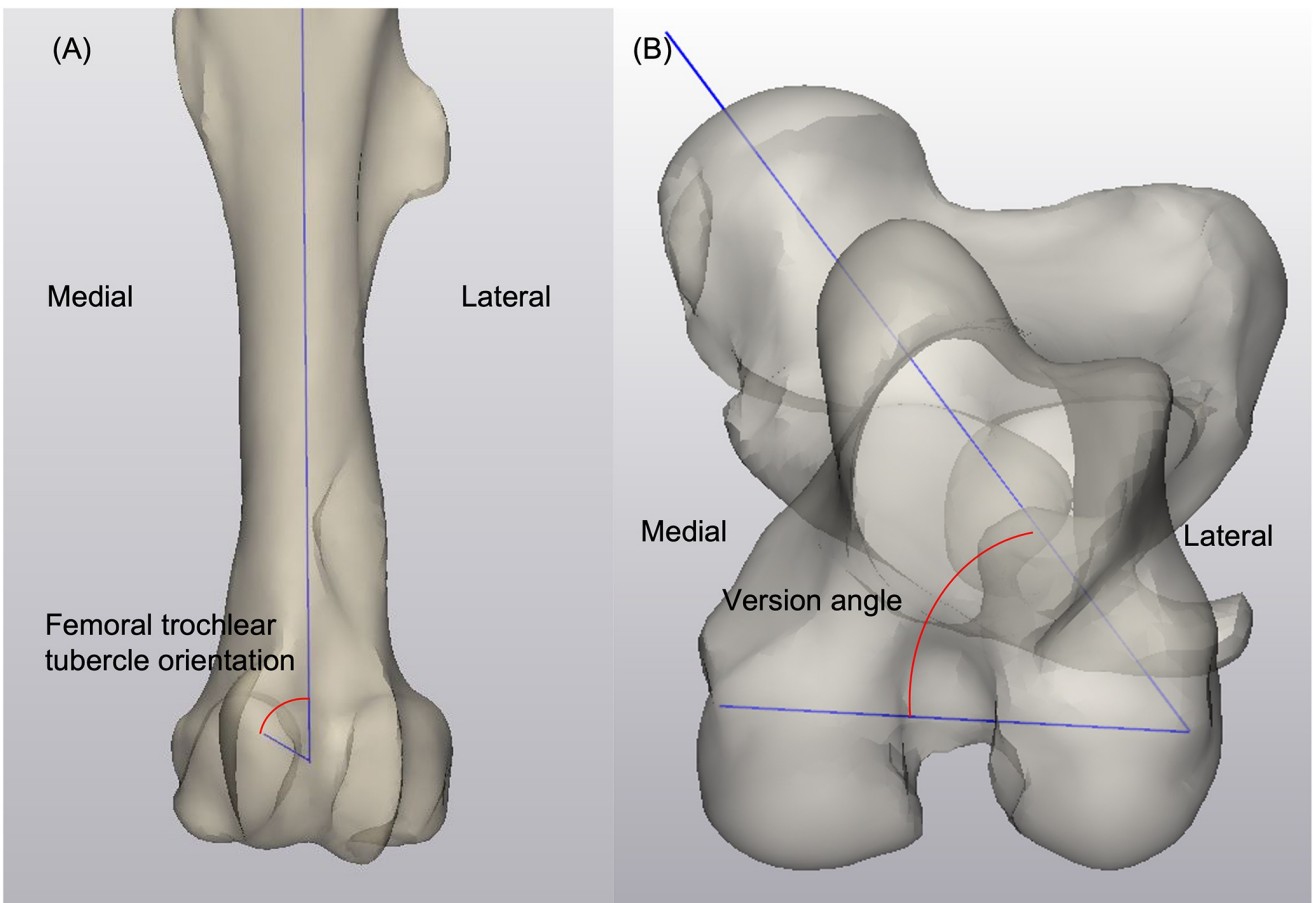

**Fig 4. Demonstration of (4A) femoral trochlear tubercle orientation on a mode 3 instance and (4B) femoral version angle on a mode 4 instance.**

### Femur shape model

In the equine femur statistical shape model (Fig 9), with 6 modes, 95.7% of the total variation was described (Table 1). Mode 1 mainly described the shape differences in scaling of all structures, which in total accounted for approximately 81.6% of the variation (Table 2). All biometrics showed strong positive correlations with the mode 1. The length of the mean femur instance was 460.6 mm and the bicondylar width was 106.7 mm. The medial condyle showed a width of 41.6 mm and a depth of 67.8 mm, respectively. Moreover, the medial condyle demonstrated a best-fit radius of 36.0 mm, and the femoral head showed a similar radius of 36.1 mm. Mode 2, which accounted for 4.8% of the total variation, demonstrated the differences in the FMA angle and the femoral neck angle (Table 3). Both biometrics demonstrated moderate correlations with the mode 2 with FMA angle being negatively correlated. Mode 3 displayed shape changes in the medial tubercle of the femoral trochlea and a moderate positive correlation with the mode 3, accounting for 3.4% of the total variation (Table 4). Mode 4 accounted for 2.5% of total variation and captured the shape differences in femoral version angle (Table 5). A weak correlation was shown between the femoral version angle and the mode 4.

### Tibia shape model

In the tibia statistical shape model (Fig 10), approximately 95% of the total shape variation within the tibia population was captured with 3 modes (Table 6). Mode 1 described the scaling

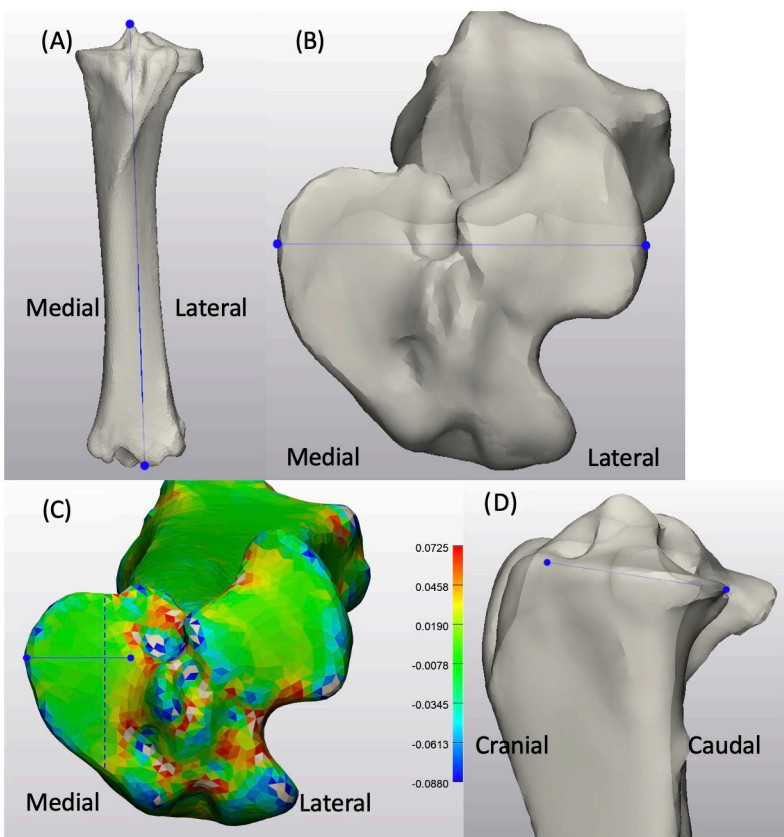

**Fig 5. Biometrics demonstrated on mode 1 instances generated by the tibia SSM model: (5A) length of the tibia, (5B) tibial bicondylar width, (5C) medial tibial condylar width, and (5D) medial tibial condylar depth.** Note that in 5C the border was defined by a Gaussian curvature analysis, where the curvature in each point is calculated based on the local minimum and maximum values using a Gaussian interpolation. The higher the curvature value, the more curved the region is. The triangle edge/point selected to be the border between the medial tibial plateau and MICET was determined by the mean +1SD curvature value, and the line (solid blue line) connecting the edge and the most medial point was perpendicular to the line directed from the most caudal to the most cranial point of the tibial plateau border defined by the curvature analysis shown as the blue dotted line.

of the tibia, accounting for 91.2% of the total variation (Table 7). All biometrics observed within the mode displayed strong positive correlations with mode 1. Mode 2 characterized the CTP angle, accounting for 2.3% of the variation (Table 8). A significantly positive strong correlation was seen between the CTP angle and the mode 2. Mode 3 described the CTS angles, which represented 1.3% of the total variation (Table 9). CTS angle in the medial compartment was moderately correlated with the mode 3 and the correlation was significant, whereas the lateral CTS angle demonstrated a non-significant weak correlation. However, there was a strong positive correlation between the medial and lateral CTS angles (R = 0.774, p < 0.001), with the lateral CTS angle being significantly larger than the medial (difference mean ± SD: 5.07˚ ± 0.37˚, p < 0.001).

## Discussion

The current study is the first report on inter-subject variability in equine femur and tibia geometry using SSM methods. The equine femur and tibia statistical shape models can be powerful tools for future studies as they provide a wide range of functionalities: facilitating

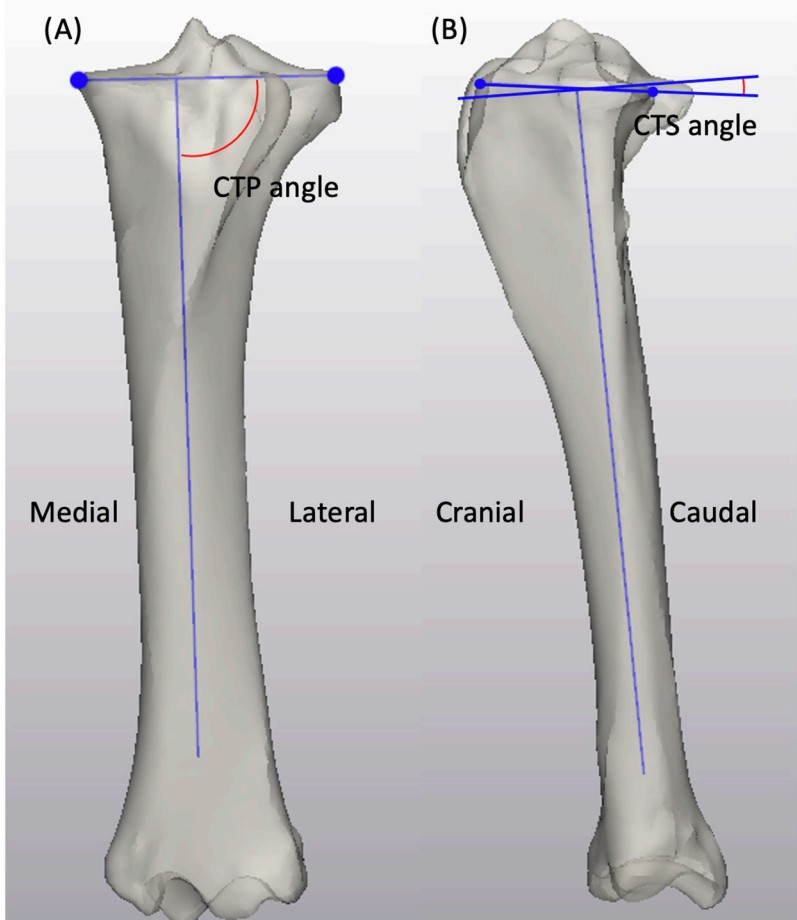

**Fig 6. Demonstration of (6A) CTP angle on a mode 2 instance and (6B) CTS angle on a mode 3 instance.**

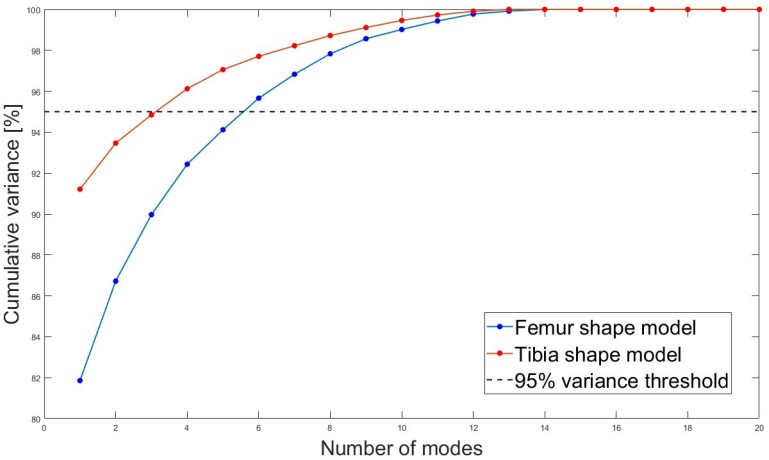

**Fig 7. Cumulative variation explained by the femur and tibia statistical shape models.**

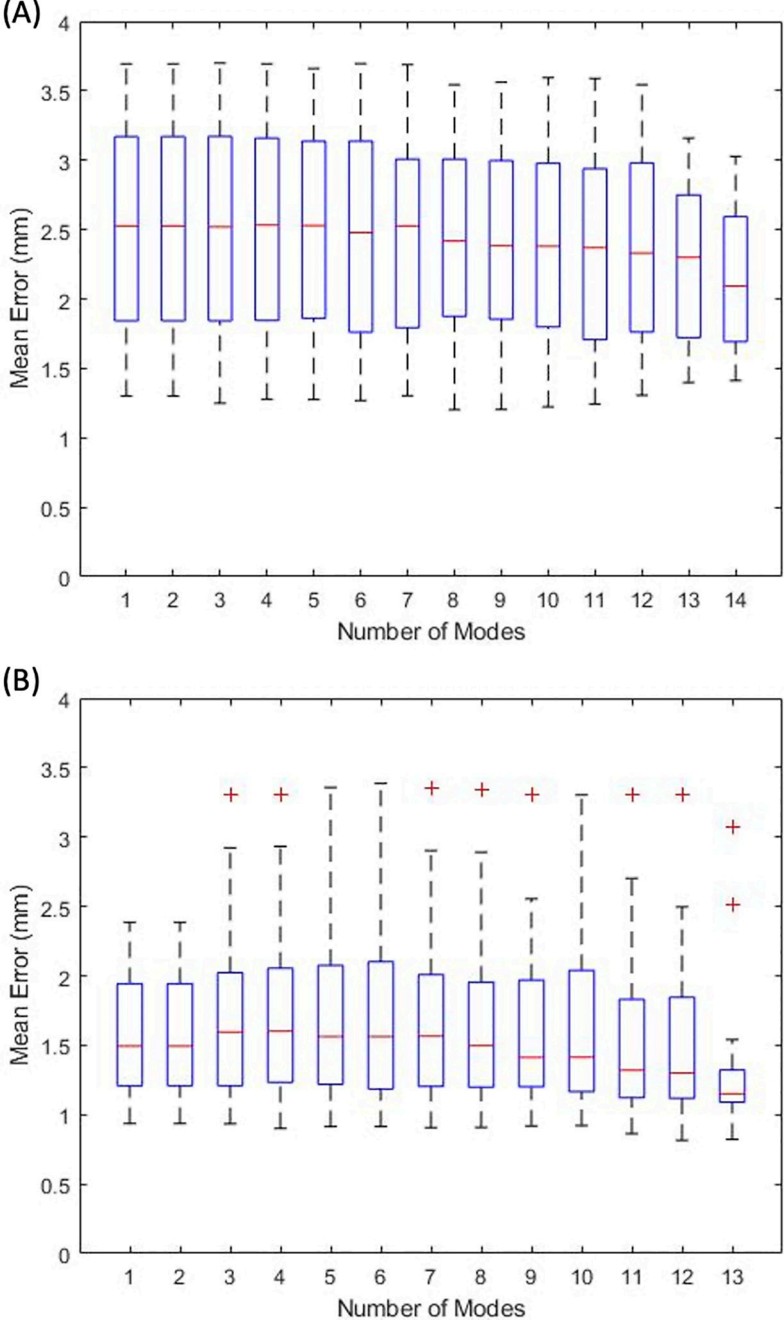

**Fig 8. Boxplot of mean geometric error evaluated in the leave-one-out analysis for (8A) femur and (8B) tibia statistical shape models as a function of the number of modes in the model prediction.** Error was calculated using the part comparison function in 3-matic (Mimics) that evaluated the mean geometric distance between the left-out model and the prediction model.

bone reconstruction, implant design and preoperative surgical planning. The statistical shape models support our hypothesis that the main mode of variation in the equine femur and tibia is due to scaling. The biometrics of the mean femur and tibia models could serve as a baseline for future references in describing average equine stifle anatomy as well as facilitating design of surgical tools and techniques such as stifle and hip arthroplasties.

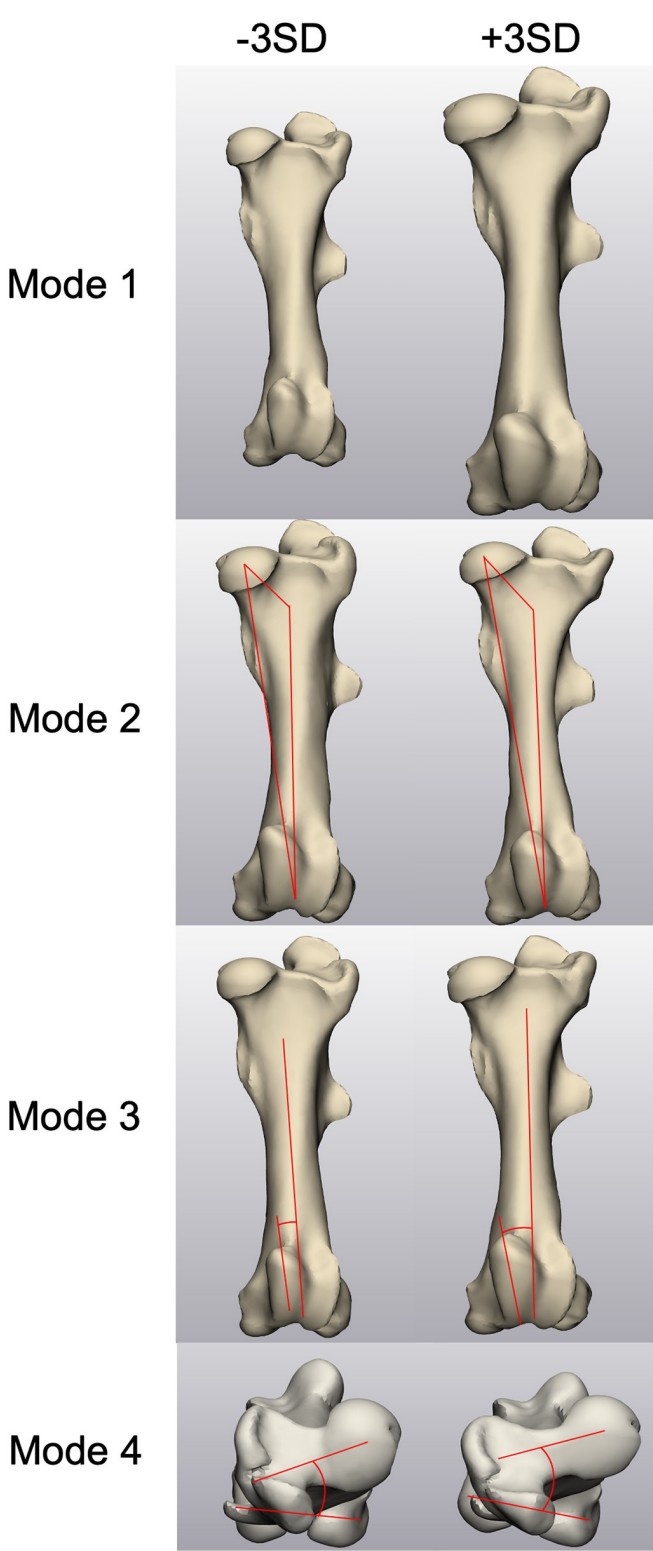

**Fig 9. Instances of the femur statistical model over ±3SD for biometrics in 4 different modes.** Note that in each graph, the measurement indicator illustrations were created to emphasize the morphology rather than showing accurate measurements.

**Table 1. Cumulative variability percentages and description of mode characteristics for the femoral shape model.**

| Mode | CUMULATIVE VARIABILITY (%) | MODE DESCRIPTION |
|---|---|---|
| 1 | 81.6 | Scaling |
| 2 | 86.4 | FMA and femoral neck angle |
| 3 | 89.8 | Femoral trochlear tubercle orientation |
| 4 | 92.2 | Femoral version angle |
| 5 | 94.1 | Third trochanter and lesser trochanter width |
| 6 | 95.7 | Third trochanter position |

**Table 2. Equine femur model measurements in mode 1.**

| Measurement | -3SD | -2SD | -1SD | MEAN | 1SD | 2SD | 3SD | R |
|---|---|---|---|---|---|---|---|---|
| Length (mm) | 400.9 | 419.5 | 439.3 | 460.6 | 476.7 | 496.5 | 515.9 | 0.993*** |
| Bicondylar width (mm) | 91.7 | 96.7 | 101.1 | 106.7 | 111.9 | 117.1 | 122.4 | 0.869*** |
| Medial condylar width (mm) | 35.8 | 38.1 | 41.3 | 41.6 | 42.8 | 45.5 | 46.0 | 0.849*** |
| Medial condylar depth (mm) | 57.1 | 60.8 | 63.8 | 67.8 | 71.3 | 74.4 | 78.1 | 0.846*** |
| Medial condylar curve (mm) | 29.5 | 31.8 | 33.1 | 36.0 | 36.3 | 38.9 | 40.3 | 0.814*** |
| Femoral head radius (mm) | 30.4 | 31.9 | 33.8 | 36.1 | 37.4 | 39.0 | 40.9 | 0.889*** |

*** P < 0.001

**Table 3. Equine femur model measurements in mode 2.**

| Measurement | -3SD | -2SD | -1SD | MEAN | 1SD | 2SD | 3SD | R |
|---|---|---|---|---|---|---|---|---|
| FMA angle (°) | 8.6 | 8.0 | 7.8 | 6.7 | 5.8 | 5.3 | 4.6 | -0.457* |
| Femoral neck angle (°) | 144.6 | 149.1 | 146.6 | 157.3 | 161.3 | 157.3 | 156.8 | 0.432*** |

* P < 0.05
*** P < 0.001

**Table 4. Equine femur model measurements in mode 3.**

| Measurement | -3SD | -2SD | -1SD | MEAN | 1SD | 2SD | 3SD | R |
|---|---|---|---|---|---|---|---|---|
| Angle between femoral trochlear tubercle and the anatomical axis (°) | 10.7 | 14.2 | 13.7 | 15.3 | 18.5 | 18.3 | 18.2 | 0.314** |

** P < 0.01

**Table 5. Equine femur model measurements in mode 4.**

| Measurement | -3SD | -2SD | -1SD | MEAN | 1SD | 2SD | 3SD | R |
|---|---|---|---|---|---|---|---|---|
| Femoral version angle (°) | 37.3 | 33.8 | 38.9 | 43.5 | 44.1 | 45.9 | 55.0 | 0.250* |

* P < 0.05

In mode 1 of the femur model, the length of the femur increased by approximately 29% from -3SD to +3SD, and the femoral condyles demonstrated a similar increase of 33% in bicondylar width. Comparably, the medial condyle showed a 28% increase in condylar width. A 37% increase in condylar depth and in condylar radius were observed in the medial femoral

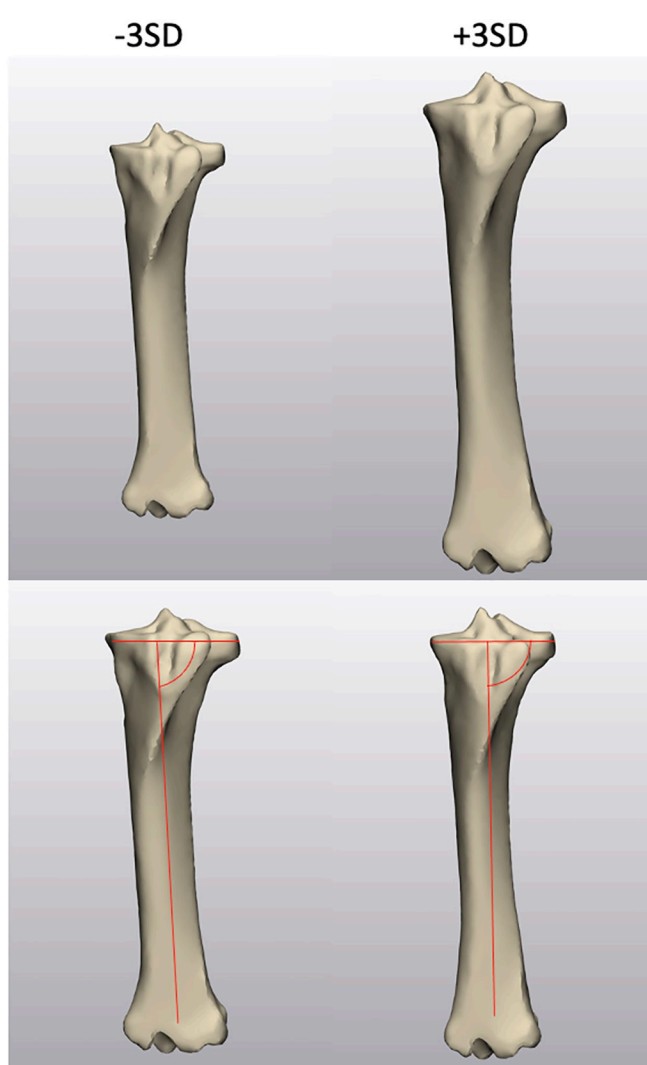

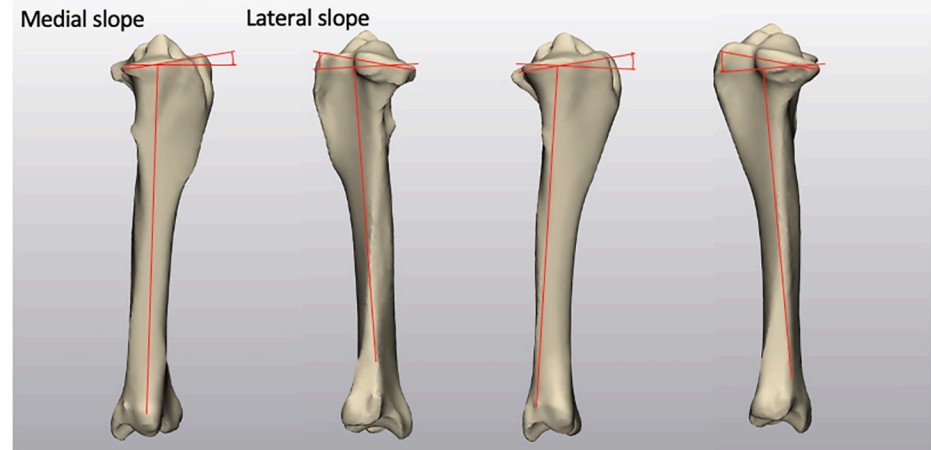

**Fig 10. Instances of the tibia statistical model over ±3SD for biometrics in 3 different modes.**

Table 6. Cumulative variability percentages and description of mode characteristics for the tibial shape model.

| Mode | CUMULATIVE VARIABILITY (%) | MODE DESCRIPTION |
|---|---|---|
| 1 | 91.2 | Scaling |
| 2 | 93.5 | CTP angle |
| 3 | 94.8 | CTS angle |

Table 7. Equine tibia model measurements in mode 1.

| Measurement | -3SD | -2SD | -1SD | MEAN | 1SD | 2SD | 3SD | R |
|---|---|---|---|---|---|---|---|---|
| Length (mm) | 358.2 | 374.8 | 392.4 | 410.3 | 427.8 | 446.8 | 464.7 | 0.998*** |
| Bicondylar width (mm) | 104.2 | 109.0 | 111.4 | 118.3 | 123.7 | 127.8 | 133.3 | 0.922*** |
| Medial condylar width (mm) | 32.6 | 35.1 | 35.1 | 35.8 | 41.5 | 44.6 | 44.9 | 0.797*** |
| Medial condylar depth (mm) | 61.6 | 72.5 | 75.8 | 79.7 | 83.9 | 86.8 | 90.3 | 0.740*** |

*** P < 0.001

Table 8. Equine tibia model measurements in mode 2.

| Measurement | -3SD | -2SD | -1SD | MEAN | 1SD | 2SD | 3SD | R |
|---|---|---|---|---|---|---|---|---|
| CTP angle (°) | 87.1 | 88.4 | 88.4 | 88.9 | 89.2 | 89.8 | 90.5 | 0.815*** |

*** P < 0.001

Table 9. Equine tibia model measurements in mode 3.

| Measurement | -3SD | -2SD | -1SD | MEAN | 1SD | 2SD | 3SD | R |
|---|---|---|---|---|---|---|---|---|
| Medial CTS angle (°) | 7.4 | 8.0 | 9.3 | 9.6 | 10.9 | 10.9 | 12.4 | 0.529* |
| Lateral CTS angle (°) | 14.4 | 15.7 | 16.3 | 17.6 | 18.6 | 19.6 | 20.9 | 0.142 |

*** P < 0.001.

condyle, demonstrating the largest proportional changes in the femur shape model. In mode 2, the femur shape model described variations in the FMA angle. Joint alignment, one of the mainstays of successful arthroplasties in human knees, requires the knowledge of the FMA angle in healthy joints, which was reported to range from 2.6° to 7.4° with a mean of 5.1° [24–27]. The current study displayed a slightly higher mean FMA angle of 6.7° ranging from 4.6° to 8.6°. In mode 3, shape variation was observed in the tubercle of the femoral trochlea. The angle between the direction of the tubercle and the anatomical axis increased by 12.4° from 6.6° to 19° within ±3SD. The result displays a wide range of trochlear positions, which could have some great impact on the position and alignment of the patella. As the direction of the tubercle aligned closer to the anatomical axis (tubercle positions more vertically), the patella would sit more laterally, which could increase the tension and stress in the medial patellar ligament resulting ligamentous tears or being more prone to injury. In mode 4, the version angle increased from 33.8° to 55° within ±3SD. In humans, the femoral neck anteversion is an essential indicator of torsion in the femur, which varies up to 30° in adults and affects the biomechanics of the knee and hip [28]. Future investigations are needed to determine the effect of different femoral version angle on the equine stifle and hip joint kinematics and kinetics.

In the tibia shape model, mode 1 accounted for 91.2% of the variation within the population. As the length of the tibia increased by approximately 30%, the bicondylar width and the medial condylar width increased by 28% and 38%, respectively. The medial condylar depth displayed a more sizable growth of 47% as the tibia length increased. These results showed that the width of the tibial plateau experienced a more similar scale growth to the tibia size, whereas the tibial condylar depth showed a higher upscaling rate than the width. Mode 2 characterizes the geometric variation in equine CTP angle. A number of studies have suggested CTP angle plays an important role in describing tibia anatomy and the joint alignment in human knees [29, 30]. In the current study, the shape model displayed a 3.4˚ variation in CTP angle with the mean instance demonstrating 1.1˚ in valgus. In mode 3, the CTS angle ranged from 7.4˚ to 12.4˚ and 14.4˚ to 20.9˚ in the medial and lateral compartments, respectively. As shown in the results, the CTS in medial and lateral compartments are positively correlated, with the lateral CTS angle being significantly larger than the medial. In human knees, the description of the posterior tibial slope was well-documented, and reports have suggested the range of the posterior tibial slope varied between races, sexes and measuring methods used [30–33]. Multiple studies have shown strong correlations between the increased posterior tibial slope and anterior cruciate ligament injury [34, 35].

In canine stifles, the CTS angle, which was reported by several groups to be 22.6˚ or greater, was much more acute than what was seen in equine stifles and human knees [36, 37]. Similarly, increased tibial slope was believed to predispose dogs to cranial cruciate ligament rupture via excessive cranial translation of the tibia [36, 38]. Therefore, future studies could utilize results derived from the current study as a baseline to determine the impact of an increased CTS on cranial cruciate ligament injury due to altered biomechanics in equine stifles.

One possible explanation for the extremely large percentage of the shape variation that mode 1 in current shape models accounted for was the size of the bones and scattered regions of interest. A common practice for modeling large bone structures with concentrated region of interest is performing customized transections on the samples, by which the investigated morphological variables would not be affected [39, 40]. However, the current study aimed to establish a comprehensive description on the femur and tibia morphology. The anatomical structures such as the diaphysis of the femur and tibia were essential to establish the anatomical and mechanical axis, thus all bone samples were intact. It is conceivable that a focus on the femoral condyles or tibial plateau would result in variation in different modes other than scaling.

One limitation to the current study is the limited sample size, which could also have contributed to the large percentage variation in mode 1 in both femur and tibia shape models. It is important to note that the modes of variation and the percentage each mode accounted for are dependent on the sampled population. The current study utilized a relatively homogenous population with 12 out of 15 specimens being Thoroughbred. Therefore, in addition to larger sample size, it would be beneficial to include more stifles from different horse breeds. Additionally, although FMA angle and CTP angle are important biometrics for investigating relationships between the joint alignment and joint disorders due to altered biomechanics, a model incorporating intact femorotibial joints would be more suited for characterizing the variability in joint alignment. Another limitation is the potential error resulted from geometric differences in the left and right equine stifles. Prior literature has shown that the mean bilateral difference in all morphological variables in canine femurs were less than 5%, thus it was assumed that errors were minimal when all right limbs were mirrored to the left in the current study [41].

The main objective of the models constructed in the current study is to provide a quantitative description of the equine femur and tibia morphological variability. As the current

imaging technology does not have the capacity to allow clinicians to CT scan an equine stifle in vivo, the models presented in the current study could possibly assist the prediction of a patient-specific bone geometry. By using different combination of modes in the current shape models, a fitting prediction geometry would be generated based on the radiographic projections. Once the matching is complete, a 3D representation of the patient's bone would be available for 3D printing, which could help clinicians to visualize or prepare for the surgical treatment.

## Conclusion

The current study presented statistical shape models of the equine femur and tibia, describing morphological variability within the sampled population in a compact representation. These population models can serve as a baseline for future studies that investigate the relationship between equine stifle morphology and joint disorders such as OA and injuries. Furthermore, development of novel stifle arthroplasties, implant design, and surgical planning could be assisted by using shape model generated instances.

## Acknowledgments

This work was supported by Surgical Translation And 3D Printing Research Lab and Gary J. Miller Ph.D. Orthopedic Biomechanics Lab at University of Florida.

## Author Contributions

**Conceptualization:** Hongjia He, Scott A. Banks, Adam H. Biedrzycki.

**Data curation:** Hongjia He, Scott A. Banks, Adam H. Biedrzycki.

**Formal analysis:** Hongjia He.

**Funding acquisition:** Adam H. Biedrzycki.

**Investigation:** Hongjia He, Scott A. Banks.

**Methodology:** Hongjia He, Scott A. Banks, Adam H. Biedrzycki.

**Project administration:** Adam H. Biedrzycki.

**Resources:** Hongjia He, Adam H. Biedrzycki.

**Software:** Hongjia He, Adam H. Biedrzycki.

**Supervision:** Scott A. Banks, Adam H. Biedrzycki.

**Visualization:** Hongjia He, Scott A. Banks, Adam H. Biedrzycki.

**Writing – original draft:** Hongjia He.

**Writing – review & editing:** Hongjia He, Scott A. Banks, Adam H. Biedrzycki.

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
