## [Decision Letter · Decision Letter 0]

23 Mar 2023

PONE-D-22-28433Anatomical variations of the equine femur and tibia using statistical shape modelingPLOS ONE

Dear Dr. Biedrzycki,

Thank you for submitting your manuscript to PLOS ONE. After careful consideration, we feel that it has merit but does not fully meet PLOS ONE’s publication criteria as it currently stands. Therefore, we invite you to submit a revised version of the manuscript that addresses the points raised during the review process. Please note that I have taken the decision to go ahead with a single review given the length of time it has taken to secure a review. Can you please address the comments raised by the reviewer?

We look forward to receiving your revised manuscript.

Kind regards,

Aliah Faisal Shaheen

Academic Editor

PLOS ONE

Journal Requirements:

2. Please provide additional information regarding the euthanasia of the animals included in the study and provide the reasons why the animals were euthanised.

3. Please include your animal ethics statement in the Methods section of your manuscript. In the Methods section of your revised manuscript, please include the animal ethics information you provided in the Cover Letter of your manuscript, the full name of the IACUC that approved the protocol, the approval or permit number that was issued, and the date that approval was granted.

“HH

The project was partially funded by the University of Florida Graduate Student Fellowship Award. The funders had no role in study design, data collection and analysis, decision to publish, or preparation of the manuscript.”

Additional Editor Comments (if provided):

Given the length of time it has taken to find reviewers for this article, I have taken the decision to go ahead with a single review. Please address the comments of the reviewer, including the rationale for developing an isolated model of the femur and tibia and technical comments related to element size.

Reviewers' comments:

Reviewer's Responses to Questions

**Comments to the Author**

1. Is the manuscript technically sound, and do the data support the conclusions?

Reviewer #1: Yes

2. Has the statistical analysis been performed appropriately and rigorously? 

Reviewer #1: Yes

3. Have the authors made all data underlying the findings in their manuscript fully available?

Reviewer #1: No

4. Is the manuscript presented in an intelligible fashion and written in standard English?

Reviewer #1: Yes

5. Review Comments to the Author

Reviewer #1: The authors present the results of a statistical shape analysis of the femur and tibia of the horse, which has been the focus of limited prior equine research.

What is the rationale for focusing on models of the femur and tibia in isolation, rather than developing a combined model that would describe full limb variation, or more specifically, stifle morphology? This would be more beneficial to the stated application of assisting with virtual surgical planning and treatment of OA.

Line 245: Exactly how was the geometric error defined/quantified? How did these errors change when they were focused around the primary region of interest (stifle) rather than the full bone (which I would expect to have very small errors along the shaft of the bone, which would bias the reported error towards smaller values)?

Fig. 8: Inclusion of additional modes seems to have a fairly small effect on the model accuracy. Why do the authors think this is? What are the leave-one-out errors if the subject-specific models were instead just compared to a (rigid) scaled version of the mean geometry? It may be that no particularly useful information (apart from size) is gained from the PCA.

“femur and tibia samples were remeshed with uniform triangles of 4.5 mm and 1.5 mm average edge lengths”. Is not clear if this means femurs were meshed with edge length of 4.5mm and tibia were meshed with edge length of 1.5mm. If this is the case, what was the difference for the meshing discrepancy between bones?

If each femoral mesh (for example) has a element size of 4.5 mm, how have the authors accounted for variability that occurs as a result of the nodal points not being directly aligned (i.e. noise) rather than true variation between bones?

Line 30-31: This is a not result and does not add value to the abstract. I suggest removing or reframing more clearly as a hypothesis.

The proposed usage for the model/data is quite vague. Clearer applications of how this work would be applied would be more helpful.

Fig. 5C. It is not clear what the reader should make of the colormap data presented in this figure. I think this represents the curvature of the mean geometry, but there is no information about how curvature changes with changing modes of variation, so it is difficult to see how this contributes to the rest of the analysis.

Lines 224-227: Why were correlation analyses limited to the relationship between CTP and medial and lateral compartments, rather than comparing the PC scores with each of the variables qualitatively identified as being representative of variation with a mode?

6. PLOS authors have the option to publish the peer review history of their article (what does this mean?). If published, this will include your full peer review and any attached files.

Reviewer #1: No

---

## [Author Response · Author response to Decision Letter 0]

8 May 2023

Response to Reviewers

Dear Editor,

Thank you for giving us the opportunity to submit a revised draft of the manuscript “Anatomical variations of the equine femur and tibia using statistical shape modeling” for publication in the PLOS ONE. We appreciate the time and effort that you and the reviewer dedicated to providing feedback on our manuscript and are grateful for the insightful comments on and valuable improvements to our paper. We have incorporated most of the suggestions made by the reviewers. Those changes are highlighted within the manuscript. A point-by-point response to the reviewers’ comments and concerns was included below.

Reviewer #1:

1. The authors present the results of a statistical shape analysis of the femur and tibia of the horse, which has been the focus of limited prior equine research. What is the rationale for focusing on models of the femur and tibia in isolation, rather than developing a combined model that would describe full limb variation, or more specifically, stifle morphology? This would be more beneficial to the stated application of assisting with virtual surgical planning and treatment of OA.

Author response: Thank you for the question. First, building a shape model of the full limb would require consistent joint pose acquired by CT scans, which can be challenging. Second, the current study aims to focus on providing a comprehensive description of the bone morphology of the equine femur and tibia. Including the full limb as a shape model would introduce pose variations, which are irrelevant to the goal of the current study.

2. Line 245: Exactly how was the geometric error defined/quantified? How did these errors change when they were focused around the primary region of interest (stifle) rather than the full bone (which I would expect to have very small errors along the shaft of the bone, which would bias the reported error towards smaller values)?

Author response: Thank you for the questions. The leave-one-out (LOO) analysis is a standard test that is commonly used for testing the generalizability of the shape model, which indicates how well the model can be generalized to new subjects. The geometric error in line 245 was referring to the LOO result, which was defined by comparing the instance generated by the prediction model with the left-out model. There are two main sources for the reported error: inaccuracy in the alignment and the model prediction. Before implementing the part comparison analysis, a global registration tool in 3-Matic was used for rigid alignment through translation and rotation. Then, the error was determined using the part comparison that calculates the mean distance between all vertices of the two models. The final result reported was the mean of the errors derived from all prediction models. Therefore, both inaccuracy in the alignment and the SSM’s prediction would contribute to geometric errors. The reviewer expected the error to be larger if the primary region of interest was the stifle rather than the full bone because of expected smaller errors along the shaft, this is not necessarily true based on what we observed from the models. In fact, most of the errors were seen at locations beyond the stifles. For example, multiple comparison showed maximum distance at the tibial malleolus and the femoral supracondylar fossa. Therefore, if the model only focused on the stifle, the overall error would be expected to be smaller.

3. Fig. 8: Inclusion of additional modes seems to have a fairly small effect on the model accuracy. Why do the authors think this is? What are the leave-one-out errors if the subject-specific models were instead just compared to a (rigid) scaled version of the mean geometry? It may be that no particularly useful information (apart from size) is gained from the PCA.

Author response: Thank you for the questions. The reason why the inclusion of additional modes did not have a huge impact on the overall accuracy is because of the large scale of the bone models. This was reflected in the variation of modes as mode 1 in the femur and tibia shape models accounted for 81.6% and 91.2%, respectively. Moreover, the percentage that the subsequent modes accounted for decreases as the number of modes increases, which means the variation in subsequent modes were much smaller in magnitude compared to scaling. 

LOO analysis was conducted to test the generalizability of the shape models using N-1 samples. So, they are essentially shape models with one fewer sample. The results showed when using the models to fit to new instances, one might expect an average geometric error of 2.12 mm and 1.37 mm when using the femur and tibia shape models with 14 and 13 PC modes, respectively. This result should be similar when the LOO model was to compare to a rigidly scaled mean geometry if the scaled model does not exceed ± 3SD in each mode. The statistical shape model describes a collection of semantically similar objects. In our case, the shape models were built using a relatively homogenous breed of horses. If the model was to predict the geometry of the femur or the tibia of a pony, it would fail to generate an accurate instance. Moreover, the comparison would likely not be meaningful as the 3D representation of a bone model is not just consist of a set of scaled biometrics.

PCA is a technique that extracts the modes of variations from a large dataset and reduces the dimensionality of the variables, which enables us to characterize complex geometries like the femur and tibia in an extremely compact manner. The biometrics reported in the current study by no means concludes all the variations encoded in each mode. Although the subsequent modes explained for much smaller percentages of the total variation compared to scaling, the morphological variations in anatomical angulation observed are still valuable and could be used for explaining joint pathology. For example, the coronal tibial slope is associated with tibiofemoral joint alignment, which is considered a major risk factor for developing osteoarthritis in human knees. Only 3.4° of difference was seen between the -3SD and +3SD instances in our model, thus there’s a homogeneity present in the equine tibia, which could explain the lack of varus/valgus joint deformity in equine stifles. Another advantage of using PCA is the ability to localize the anatomical differences. With every mode of variation being independent of each other, we can identify the shape changes that each mode corresponds to through manipulating the standard deviation.

4. “femur and tibia samples were remeshed with uniform triangles of 4.5 mm and 1.5 mm average edge lengths”. Is not clear if this means femurs were meshed with edge length of 4.5mm and tibia were meshed with edge length of 1.5mm. If this is the case, what was the difference for the meshing discrepancy between bones? If each femoral mesh (for example) has an element size of 4.5 mm, how have the authors accounted for variability that occurs as a result of the nodal points not being directly aligned (i.e. noise) rather than true variation between bones?

Author response: Thank you for pointing this out. The texts have been revised in materials and methods line 125:

 “…femur and tibia samples were remeshed with respective uniform triangles of 4.5 mm and 1.5 mm average edge lengths.”

As the femurs have much larger surface than the tibiae, more triangles are required to accurately represent the geometry of the femurs. The larger the edge length of the mesh is, the fewer triangles we need for meshing. Literature on statistical shape model of the full human femur had used approximately 8,000 nodes to identify the geometric variations. Therefore, we aimed for a similar number of nodes on the femur and tibia samples. As mentioned in materials and methods line 130, 4.5 mm edge length for the femur mesh and 1.5 mm for the tibia mesh were used and yielded approximately 14,000 to 18,000 (approximately 8000 nodes) and 11,000 to 14,000 (approximately 7000 nodes) triangles for femora and tibiae, respectively. 

Multiple steps were carried out after remeshing to minimize the geometric errors resulted from alignment. All samples were rigidly aligned with scaling to the template through 50 iterations using 75% of the sample points after remeshing. As for the warp errors, also referred as elastic registration errors in some literature, they were negligibly small and often were not included. In our model, the unsigned errors were 0.184 ± 0.061 mm and 0.151 ± 0.045 mm in the femur and tibia shape models, respectively. Finally, the Procrustes alignment method was used (integrated in create SSM function in 3-Matic) before PCA to eliminate position differences and ensure only shape variations were captured.

The texts from 131 to 136 were revised to clarify the preprocess preparing for the PCA:

“Alignment with scaling of the samples were carried out with each sample being automatically rigidly aligned to the template and scaled. To minimize the computational expense while maintaining good mesh fidelity, sample points were set at 75% with 50 iterations for each alignment. A warp function was used to create point correspondence between the template and the sample after the alignment. Procrustes alignment was carried out to eliminate the position differences and ensure only shape variations were captured.”

5. Line 30-31: This is a not result and does not add value to the abstract. I suggest removing or reframing more clearly as a hypothesis.

Author response: Thank you for the suggestion. We absolutely agree and the text has been removed as suggested. 

6. The proposed usage for the model/data is quite vague. Clearer applications of how this work would be applied would be more helpful.

Author response: Thank you for the suggestion. Currently, the model was used to generated multiple instances for facilitating designing implants and surgical instruments for an unicompartmental equine stifle arthroplasty. Moreover, the model can be used in assisting equine anatomy education. As discussed in line 396 to 400, for the use of the patient-specific bone matching, prediction model can be generated using different combination of modes in statistical shape model. Then the prediction geometry will be used to match the radiograph of the patient. Once matching is complete, the SSM can export the 3D model of the prediction geometry for clinicians to observe and practice on. 

Lines from 397 to 400 were revised as followed:

“the models presented in the current study could possibly assist the prediction of a patient-specific bone geometry. By using different combination of modes in the current shape models, a fitting prediction geometry would be generated based on the radiographic projections. Once the matching is complete, a 3D representation of the patient’s bone would be available for 3D printing, which could help clinicians to visualize or prepare for the surgical treatment. ”

Another helpful application is the model can serve as a representation of the healthy equine stifle population. Future studies can build a shape model using stifles with joint pathologies. By comparing the healthy model and the joint model with pathologies, this could help identify the abnormal anatomical variations that contribute to the pathologies. 

7. Fig. 5C. It is not clear what the reader should make of the colormap data presented in this figure. I think this represents the curvature of the mean geometry, but there is no information about how curvature changes with changing modes of variation, so it is difficult to see how this contributes to the rest of the analysis.

Author response: Thank you for your question. The colormap in Fig 5C indeed represents the curvature of the medial tibial plateau of the mean geometry. As there is no clear definition where the medial tibial plateau ends, it is challenging to define the width of the medial tibial plateau manually without introducing human errors. Therefore, we decided to use the curvature as an indicator with an automated algorithm to consistently separate the plateau and the MICET. Because the curvature analysis was only used to identify the border of the medial plateau, the magnitude change of the curvature was irrelevant. Furthermore, all curvature analysis was carried out on mode 1 instances as mode 1 characterizes the scaling of the condylar dimensions. The use of the curvature analysis for determining the width of the medial tibial plateau was stated in line 201 to 203. 

8. Lines 224-227: Why were correlation analyses limited to the relationship between CTP and medial and lateral compartments, rather than comparing the PC scores with each of the variables qualitatively identified as being representative of variation with a mode?

Author response: Thank you for your question. First of all, we want to apologize for the typo in statistics line 247, the correlation analysis was performed on the CTS in the medial and lateral compartments as shown in the results instead of the CTP. The text has been revised in the manuscript. 

Second, the initial rationale for only including the CTS correlation was to emphasize the interest on tibial slope comparison as it’s an important biometric associated with joint stability and joint kinematic performance in many species. However, the authors agree that it’s a great suggestion to add correlation analysis between the PC score and each biometrics. Thus, we generated 100 arbitrary femur and 100 tibia instances using the shape models with randomized standard deviation in each mode, the biometrics in each instance were measured using a script in MATLAB. Pearson’s correlation coefficient R and the p-value were calculated all analyses. An additional Pearson’s correlation analysis and a paired T test were performed to evaluate the relationships between the medial and lateral CTS angles. The manuscript in materials and methods statistics line 246 to 254 and in results were revised.

---

## [Decision Letter · Decision Letter 1]

5 Jun 2023

Anatomical variations of the equine femur and tibia using statistical shape modeling

PONE-D-22-28433R1

Dear Dr. Biedrzycki,

We’re pleased to inform you that your manuscript has been judged scientifically suitable for publication and will be formally accepted for publication once it meets all outstanding technical requirements.

Kind regards,

Aliah Faisal Shaheen

Academic Editor

PLOS ONE

Additional Editor Comments (optional):

Thank you for addressing the reviewers' comments. The manuscript is now accepted for publication.

Reviewers' comments:

Reviewer's Responses to Questions

**Comments to the Author**

1. If the authors have adequately addressed your comments raised in a previous round of review and you feel that this manuscript is now acceptable for publication, you may indicate that here to bypass the “Comments to the Author” section, enter your conflict of interest statement in the “Confidential to Editor” section, and submit your "Accept" recommendation.

Reviewer #1: All comments have been addressed

2. Is the manuscript technically sound, and do the data support the conclusions?

Reviewer #1: (No Response)

3. Has the statistical analysis been performed appropriately and rigorously? 

Reviewer #1: (No Response)

4. Have the authors made all data underlying the findings in their manuscript fully available?

Reviewer #1: (No Response)

5. Is the manuscript presented in an intelligible fashion and written in standard English?

Reviewer #1: (No Response)

6. Review Comments to the Author

Reviewer #1: (No Response)

7. PLOS authors have the option to publish the peer review history of their article (what does this mean?). If published, this will include your full peer review and any attached files.

Reviewer #1: No

---

## [Editor Report · Acceptance letter]

22 Jun 2023

PONE-D-22-28433R1 

Anatomical variations of the equine femur and tibia using statistical shape modeling 

Dear Dr. Biedrzycki:

I'm pleased to inform you that your manuscript has been deemed suitable for publication in PLOS ONE. Congratulations! Your manuscript is now with our production department. 

Kind regards, 

on behalf of

Dr. Aliah Faisal Shaheen 

Academic Editor

PLOS ONE